# *VvPL15* Is the Core Member of the Pectate Lyase Gene Family Involved in Grape Berries Ripening and Softening

**DOI:** 10.3390/ijms24119318

**Published:** 2023-05-26

**Authors:** Yuying Ma, Chukun Wang, Zhen Gao, Yuxin Yao, Hui Kang, Yuanpeng Du

**Affiliations:** State Key Laboratory of Crop Biology, Collaborative Innovation Center of Fruit & Vegetable Quality and Efficient Production in Shandong, College of Horticulture Science and Engineering, Shandong Agricultural University, Taian 271018, China

**Keywords:** grape, softening, pectate lyase (PL), pectin

## Abstract

The process of ripening and softening in grape begins at veraison and is closely related to the depolymerization of pectin components. A variety of enzymes are involved in pectin metabolism and one class of enzyme, pectin lyases (PLs), have been reported to play an important role in softening in many fruits; however, little information is available on the *VvPL* gene family in grape. In this study, 16 *VvPL* genes were identified in the grape genome using bioinformatics methods. Among them, *VvPL5*, *VvPL9*, and *VvPL15* had the highest expression levels during grape ripening, which suggests that these genes are involved in grape ripening and softening. Furthermore, overexpression of *VvPL15* affects the contents of water-soluble pectin (WSP) and acid-soluble pectin (ASP) in the leaves of Arabidopsis and significantly changes the growth of Arabidopsis plants. The relationship between *VvPL15* and pectin content was further determined by antisense expression of *VvPL15*. In addition, we also studied the effect of *VvPL15* on fruit in transgenic tomato plants, which showed that *VvPL15* accelerated fruit ripening and softening. Our results indicate that *VvPL15* plays an important role in grape berry softening during ripening by depolymerizing pectin.

## 1. Introduction

Grape (*Vitis vinifera* L.) is one of the most economically important fruit crops in the world and is also an important genetic resource for woody fruit crops [1]. Grapes are widely popular because of their nutritional value and health benefits. However, fruit softening after ripening is a major contributor to postharvest deterioration, which directly affects storage logistics and shelf life [2]. Therefore, identifying fruit softening-specific regulatory genes, balancing texture and other fruit quality traits, and obtaining germplasm varieties that show delayed softening while maintaining good fruit quality have become the goal of grape breeders and scientists [3].

Fruit softening is an important characteristic of fruit ripening, which occurs at veraison (the onset of ripening) and mainly involves significant changes in the properties of cell wall polysaccharides [4] and turgor pressure [5,6]. The structural changes that occur during fruit softening are mainly caused by changes in the mechanical strength of the cell wall and intercellular adhesion, and pectin depolymerization is considered to be one of the most important factors that causes fruit softening in different species [7]. The plant cell wall is composed of the middle lamella, the primary cell wall, and the secondary cell wall. The primary cell wall consists of pectin, hemicellulose, and cellulose [8]. Pectin is an important component of the cell wall in higher plants, and it is mainly present in the primary wall and middle layer [9]. It is a complex polysaccharide that is mainly composed of three domains, including Homo-galacturonan (HG), Rhamno-galacturonan I (RG-I), and Rhamno-galacturonan II (RG-II) [10]. Pectin is involved in the adhesion between cell wall cells mainly through Ca^2+^ cross-linking with partially de-esterified galacturonic acid glycans in the middle layer [11]. In general, the main causes of fruit softening are cell wall disassembly and reduction in the cell-to-cell adhesion, which is the result of middle lamella dissolution [12]. While the dissolution of pectin is one of the most significant changes that occurs during fruit softening, with the dissolution of the pectin layer, the intercellular space becomes larger and the fruit softening intensifies. During fruit ripening and softening, the changes in pectin in the intercellular layer and the cell wall mainly include the degradation of insoluble protopectin into soluble pectin, the removal of methyl groups in polymers, and the decrease in the contents of neutral sugars, such as galactose and arabinose [13].

Pectin metabolism in plants involves a variety of enzymes, including pectin methylesterase (PME, EC3.1.1.11), polygalacturonase (PG, EC3.2.1.15), and pectin lyase (PL, EC4.2.2.2) [14,15,16]. In particular, pectate lyase (PL) is a key component of the enzyme mixture involved in maturation-related cell wall remodeling, and it plays an important role in the softening process of succulent fruits [17]. A study of transgenic lines of strawberry (*Fragaria* × *ananassa*) expressing antisense *FaPL* (*njjs25*) provided the first experimental evidence of the role of PL in fruit firmness [18]. The high expression levels of the *MaPel I* and *MaPel II* genes during fruit ripening has also been reported in banana [19]. *PpePL1* and *PpePL15* are involved in fruit ripening and softening in peach [20], and *VvPL11* plays a role in ripening and softening in grape berries [21]. Recently, the mechanism of action of PL has been studied in detail; for example, silencing *SlPL* inhibited fruit softening and reduced the water soluble pectin (WSP) content in tomato [22], and transcription factors, such as SlLOB1 and SlERF.F12 can play a role in fruit ripening by regulating expression of *SlPL* [23,24]. In strawberry fruits, FvPL appears to act against tightly bound pectin [25], and FvWRKY48 binds to the pectate lyase *FvPLA* gene promoter to control fruit softening in *Fragaria vesca* [26]. However, the function of *PL* in grape fruits is still less reported.

In this study, we identified genes encoding PL family members that are expressed during ripening and softening in grape berries. We analyzed and predicted PL family members using bioinformatics methods, and the results provide valuable clues to predict their function. Genetic transformation and virus-induced gene silencing technology (VIGS) were used to clarify the function of *VvPL15* in grape berry softening. Our research provides a foundation for further investigation of the function of PL family members in the process of ripening and softening in grape berries.

## 2. Results

### 2.1. Determination of Pectin Content in ‘Kyoho’ Grape Fruit Flesh during Ripening

Veraison is a critical period in the grape life cycle as it signals the onset of fruit ripening. We collected ‘Kyoho’ grape berries at five stages of development. The skin color began to change at E-L 35 and gradually became darker, indicating that E-L 35 is the onset of veraison (Figure 1A). Moreover, we found that grape flesh firmness decreased rapidly from E-L 35 to E-L 36, after which the rate slowed (Figure 1B). At the same time, the WSP content increased gradually (Figure 1C), but the ASP content decreased continuously (Figure 1D). These results indicated that the firmness of grape berries could be related to the changes in pectin components.

### 2.2. Identification of VvPLs in Grape

A total of 16 genes were identified from the grapevine genome through orthologues analysis with *Arabidopisis thaliana* and *Solanum lycopersicum*. These genes were designated as *VvPL1-VvPL16* for VvPLs according to their corresponding location on each chromosome (from top to bottom). A brief summary of basic information for VvPLs in grapevine is listed in Table 1, where the basic features include the corresponding gene names, gene length, chromosome location, protein properties and subcellular localization. In general, the gene length of VvPLs varied from 1581 to 7549 bp, while the amino acid length ranged from 320 to 496 amino acid (aa), respectively. Moreover, the molecular weight (kDa) ranged from 36.1 to 53.9 for VvPLs, while the theoretical isoelectric points (pIs) varied from 4.98 to 9.66 for VvPLs. Furthermore, *VvPL* family members were unevenly distributed among the ten chromosomes of the grape genome and three, two and one members were distributed on each chromosome (Figure 2). The tandem duplication sites (*VvPL1* and *2, VvPL9* and *10*) were found on chromosomes 1 and 13. In addition, according to web-based prediction of *VvPLs* location in cells, different members of *VvPL* gene family were mainly found in the cell wall.

### 2.3. Phylogenetics Analysis of VvPLs

The phylogenetic relationship of 16 VvPLs of grape and three other plants were obtained using MEGA7.0 via the neighbor-joining approach. Based on the classification of AtPLLs and SlPLs, the phylogenetic tree revealed that VvPLs and PLs of three other plants evolved from a common ancestor and VvPLs can further be divided into five major groups (Figure 3). In phylogenetic tree of VvPLs, we observed that group 1 contained the most number of genes compared to group 2 and 3 and consisted of four members of the VvPLs family. Meanwhile, VvPL7 and VvPL13 were clustered into group 4, in which VvPL13 showed a closer relationship with AtPLL3. In addition, VvPL8 was only one member of the VvPLs family in group 5, suggesting a specific function of VvPLs in grape.

### 2.4. Protein and Gene Structure Analysis of VvPLs

The phylogenetic tree among VvPLs showed VvPL proteins are divided into five major groups due to variations in tree topologies (Figure 4A). We analyzed the composition of motifs for VvPLs and obtained ten conserved motifs using the online server, Multiple Em for Motif Elicitation (MEME). The results revealed that the motifs six, two and one were dominantly found among PL members and 9 PL members contained ten motifs except members 16, 9, 8, 13 and 7 (Figure 4B). These results suggest that proteins on the same branches may have broadly similar functions. Furthermore, based on the coding sequence (CDS) and untranslated regions (UTR) of *VvPL* genes in grape, gene structures were resolved using the online tool GSDS2.0. The results revealed that *VvPL* members exhibited high divergence and largely conserved compared to each other, again suggesting that genes from the same branch may have retained similar functions during plant evolution (Figure 4C). In addition, 3D models of the VvPL proteins were created using Phyre. The individual 3D models indicated that VvPLs also shared a common structure (Figure 5).

### 2.5. Collinearity Analysis of the VvPLs

To reveal the expansion mechanism of the *VvPLs* family, intergenomic duplication data filers of grape and Arabidppsis were filtered using TBtools. Microsynteny between species can be used to identify the location of orthologous genes. In total, we identified 23 orthologous gene pairs between grape and Arabidppsis (Figure 6). This indicates that numerous grape and Arabidppsis counterparts may have evolved from a common ancestor. This result suggests that the grape orthologous genes functions may be predicted by the Arabidppsis orthologous genes.

### 2.6. Cis-Regulatory Element Analysis in VvPL Genes

The cis-acting elements present in the *VvPL* gene promoters were analyzed using the online PlantCARE database. The number of cis-elements identified in the promoter regions of the *VvPL* genes ranged from 6 to 26 (Appendix A). Our analysis included hormone-related responsive elements, such as those for abscisic acid (ABA), gibberellin (GA), auxin, salicylic acid (SA), methyl jasmonate (MeJA), and ethylene [27]; stress-related responsive elements, such as those for low temperature and drought; and also typical binding sites for transcription factors from families, such as MBS and W-box (Appendix A) [28,29]. We also used TBtools software to construct distribution maps of the promoter sequences showing the number and positions of the cis-elements (Figure 7). A total 16 members of the *VvPL* family were analyzed, among which 14 of the genes had ERE elements and 11 genes had ARBEs (Appendix A). Almost all of the gene promoters contained multiple MYB elements, and most of the promoters contained TCA-elements, CGTCA-motifs, and W-box elements.

### 2.7. Identification of VvPL Gene Family Members Related to Fruit Flesh Softening

To characterize the putative function of *VvPL* members, we examined the expression levels of 16 *VvPL* genes via RT-qPCR during ripening and softening in ‘Kyoho’ grape berries. The results showed that all the 16 members were expressed, but only 6 members (*VvPL3*, *VvPL4*, *VvPL5*, *VvPL9*, *VvPL13*, and *VvPL15*) showed upregulated transcription at E-L 35. The expression of three genes (*VvPL5*, *VvPL9*, and *VvPL15*) changed significantly from E-L 35 to E-L 38 (Figure 8B), and *VvPL15* showed the highest relative expression level during grape berry ripening and softening (Figure 8A,B). Therefore, it is inferred that *VvPL15* (*VIT_217s0000g09810*) may play an important role in the process of ripening and softening in grape berries.

### 2.8. VvPL15 Overexpression Affects Pectin Content and Plant Growth in Transgenic Arabidopsis

To further study the effect of *VvPL15* on pectin content, a vector was constructed to overexpress *VvPL15* in transgenic Arabidopsis Col-0 plants. To verify that the transgenic Arabidopsis plants carried the *VvPL15*-overexpression cassette, DNA was extracted from mature rosette leaves for PCR *VvPL15* gene amplification. DNA fragments with lengths identical to that of the *VvPL15* coding region were amplified (Appendix A), indicating that *VvPL15* had been transferred into the Arabidopsis genome and was successfully expressed in T_0_-generation plants. Three transgenic lines (*VvPL15-OX #2*, *VvPL15-OX #4*, and *VvPL15-OX #6*) were selected for phenotypic analysis. The results showed that the leaves of the overexpression lines were significantly smaller than those of the wild type and had a curly shape (Figure 9A). In addition, the overexpression lines were significantly thinner and shorter (Appendix A). We next measured the pectin content of Arabidopsis leaves. The results showed that compared with the wild-type, the WSP content in the Arabidopsis overexpression leaves increased significantly, while the ASP content decreased (Figure 9B,C). These results show that overexpression of *VvPL15* affects the growth of Arabidopsis plants, which may be related to pectin degradation.

### 2.9. Silencing VvPL15 Expression Affects the WSP and ASP Contents in Grape Leaves

To further investigate the role of *VvPL15* in grape, we performed both VIGS and transient overexpression of *VvPL15* in grape leaves. Softening phenotypes were analyzed, and pectin content in antisense expression and overexpressed grape leaves were studied in detail (Figure 10). VvPL15-TRV infiltration was the RNAi treatment in leaves, and TRV infiltration was the control. VvPL15-PRI infiltration was used to overexpress *VvPL15* in the leaves, and PRI infiltration was used as the control. *VvPL15* expression was inhibited in VvPL15-TRV grape leaves compared with the control. In VvPL15-PRI grape leaves, *VvPL15* expression was significantly up-regulated (Figure 10A). The WSP content in VvPL15-TRV leaves was lower than in the control, while the ASP content was significantly higher than in the control (Figure 10B). On the contrary, the WSP content of VvPL15-PRI leaves was lower than it was in the control, and the ASP was higher than in the control (Figure 10C). These experiments demonstrate that pectin degradation was delayed after *VvPL15* expression was downregulated, and that pectin degradation was accelerated by *VvPL15* overexpression.

### 2.10. VvPL15 Overexpression Accelerates Fruit Softening in Tomato

To elucidate the function of *VvPL15* during fruit ripening, we overexpressed *VvPL15* in tomato plants (Figure 11). We obtained three transgenic tomato lines overexpressing *VvPL15* (*VvPL15-OX # 1*, *VvPL15-OX #2*, and *VvPL15-OX #3*) and showed that they were transgenic via PCR (Appendix A). Under suitable growth conditions, there was a significant difference between the *VvPL15*-overexpressing (OX) lines and the wild-type (WT) at the seedling stage. The seedlings of the *VvPL15*-OX lines showed weaker growth compared to WT plants (Figure 11A), which was the same as we observed in the Arabidopsis Col-0 plants overexpressing *VvPL15* (Appendix A). In addition, the changes in the characteristics of the fruits of the transgenic lines were also obvious. We observed that the fruits of the *VvPL15*-OX tomato lines turned red earlier than they did on WT plants, suggesting that the breaker stage (BR) occurred earlier in the transgenic tomato fruits (Figure 11B). The firmness of the transgenic tomato fruits was decreased significantly compared with WT fruits (Figure 11C). These results indicate that overexpression of *VvPL15* in tomato can accelerate tomato fruit ripening and softening. Furthermore, we analyzed the changes in fruit pectin content. In the transgenic tomato fruits, the ASP content was greatly reduced (Figure 11E), and the WSP content was higher in all stages and all three OX lines compared with the WT (Figure 11D). The firmness and pectin content showed the largest change in MG and BR. Therefore, overexpression of *VvPL15* in tomato results in tomato fruits that mature earlier. Furthermore, overexpression of *VvPL15* promoted fruit softening, which might be attributed to the degradation of ASP.

## 3. Discussion

### 3.1. The VvPL Gene Family and Its Evolutionary Analysis in Grape

The first gene encoding a PL enzyme (also known as Pectate lyase-like, *PLL*, or *PEL*) in higher plants was cloned from tomato pollen [16]; in our study, we identified 16 *VvPL* family members in the grape genome. *PL* genes have been found in other plants; for example, there are 26 *AtPLL* genes in Arabidopsis [30], 12 *OsPLL* genes in rice [31], 46 *BrPLL* genes in *Brassica rapa* [32], and 30 *PtPL1* genes in the *Populus trichocarpa* genome [33]. There are also 20 *PpePL* genes in peach and 22 *SlPL* genes in the tomato genome [20,22], and 53, 42, and 83 PEL family genes have been identified in the genomes of three cotton species, *Gossypium raimondii*, *G. arboretum*, and *G. hirsutum*, respectively [34]. The reason for the small number of grape *VvPL* family genes could be that they did not undergo genome-wide duplication during the evolution of *V. vinifera*. Gene duplication is considered to be one of the main drivers of evolution and the expansion of gene families, and segmental duplication and tandem duplication are two main mechanisms that can cause the expansion of plant gene families [35,36]. For example, *PpePL5*, *6*, *7*, and *8* have been identified as arising from tandem repeats, which may have resulted in the expansion of the *PpePL* gene family, leading to changes in the structure and function of family members [20]. This is consistent with previous studies on tomato [22], poplar [32], and cotton [34]. The PL phylogenetic tree were clustered into five groups. Moreover, a multiple protein sequence alignment showed that all plant PL proteins contain a conserved Pec_Lyase_C domain, and that some PL proteins have a signal peptide and a Pec_Lyase_N domain located at the N-terminus [37]. This indicates that PL proteins are highly evolutionarily elements conserved in plants. In this study, the conserved nature of the *VvPL* gene family was further demonstrated via the collinearity analysis of *VvPL* genes from the grape and Arabidopsis genomes (Figure 6). These research results indicate that the evolution of *PL* gene family in *V. vinifera* is very conservative.

### 3.2. Functional Analysis of the VvPL Gene Family in Grape

Analysis of *VvPL* gene promoter sequences showed that there are many types of cis-elements present that allow gene expression to respond to different environmental stimuli, and this may be closely related to the multiple functions of *VvPL* genes in plant species. A total of 16 members of the *VvPL* family were identified, among which 14 gene promoters had ERE elements and 11 had ARBEs (Appendix A). These results indicate that members of this gene family could play roles in grape fruit ripening and abscission. Almost all of the *VvPL* gene promoters were found to contain multiple MYB elements, suggesting that the transcription of these genes may be regulated by MYB transcription factors. Most of the promoters contain TCA-elements, CGTCA-motifs, and w-box elements, suggesting that the gene family may be involved in plant defense responses. In addition, many known *PL* genes involved in growth and development were included in the phylogenetic tree. In group 1, *VvPL15* showed a close relationship with *SlPL*, which encodes an important pectinase associated with tomato cell wall decomposition and fruit softening [22]. This indicates that *VvPL15* may play a similar role in regulating grape softening. In the first group, *VvPL5* is highly homologous to *OsPLL4*, and *OsPLL4* is associated with pollen and grain development [31]. *VvPL9* belongs to group 3 and is on the same branch with *AtPLL12*. *AtPLL12* is expressed in the phloem and promotes the development of the cambium and xylem, which is necessary for the growth of vascular bundles in Arabidopsis inflorescence stems [38]. This suggests that *VvPL9* may be involved in cell expansion. Moreover, in our study, transgenic Arabidopsis and tomato plants have shown that *VvPL15* affects plant growth (Figure 9 and Figure 11). Therefore, *VvPL15* may have divers functions, which still need further study.

### 3.3. VvPL15 Is a Key PL Family Member Associated with Fruit Softening in Grape

Ripening and softening of fruits is the main factor that determines their shelf life and commercial value. Many studies have shown that plant PL proteins participate in the ripening and softening process in fruits by degrading demethylated pectin in cell walls [10]. *PL* genes have been shown to be specifically expressed during fruit ripening, such as *Pm65* (pectin lyase) in Chinese plum (*Prunus mume*) [39], banana (*Musa acuminata*) [40,41], and mango (*Mangifera indica*) [42]. Our study showed that *VvPL15* is highly expressed during stages E-L35 and E-L36 in grape flesh (Figure 8), and that overexpression of *VvPL15* in transgenic tomato accelerated ripening, which suggests that *VvPL15* may play an important role during ripening and softening in grape berries. The content of ASP in transgenic tomato fruit decreased and the content of WSP increased, showing that degradation of protopectin accelerates fruit softening and texture changes. These results are also consistent with those of previous studies in species, such as strawberry [18,43], peach [20], and tomato [22]. Recently, Li et al. identified 18 *VvPL* genes and found that *VvPL11* (*VIT_213s0019g04910*) is a key member of the pectin lyase gene family associated with fruit softening [21]. Compared with the study of Li et al., we identified two fewer *VvPL* genes in our study. The reason is that the extremely short sequences of *VvPL4* (*VIT_201s0010g03340*) and *VvPL18* (*VIT_200s0317g00150*) located on chromosome 0 were excluded in our analysis. In addition, differences were found in the key genes involved in fruit softening. We found that *VvPL15* (*VIT_217s0000g09810*) is a key gene involved in fruit ripening and softening in ‘Kyoho’ grape, whereas Li et al. found that *VvPL11* (*VIT_213s0019g04910*) is a key gene that affects fruit softening in ‘Hanxiangmi’ grape. Therefore, *VvPL* family members that play key roles in fruit ripening and softening may differ between grape varieties.

## 4. Materials and Methods

### 4.1. Plant Materials and Growth Conditions

Four-year-old rooted cuttings of grapevine (*Vitis vinifera* cv. ‘Kyoho’) grown under standard field conditions were selected from the Shandong Agricultural University Horticultural Experiment Station (Taian, China). The berries were collected at five developmental stages; E-L 33, E-L 35, E-L 36, E-L 37, and E-L 38 based on the standard system for classifying grapevine growth stages (E-L system). The grape flesh firmness was determined immediately after sampling, and the samples were then frozen in liquid nitrogen and stored at −80 ℃ prior to their use in the experiments.

*Arabidopsis thaliana* ecotype Columbia-0 (Col-0) was used as the wild-type. Seeds of Col-0 were sterilized with 70% alcohol and 1% sodium hypochlorite (NaClO) and sown on 0.5X MS semisolid medium (supplemented with 8 g·L^−1^ agar and 30 g·L^−1^ sucrose). Seeds were vernalized by incubation at 4 °C for three days and then transferred to a growth chamber under a 16 h/8 h (light/dark) cycle (200 μmol·m^−2^·s^−1^) at 21 °C/18 °C (day/night) for nine days. Seedings were then cultivated in nutrient soil and grown in agreenhouse under the same conditions as in the growth chamber.

*Solanum lycopersicum* ‘Ailsa Craig’ (AC) was used in transformation experiments. Transgenic cultures were grown in a greenhouse under a 16 h/8 h (light/dark) photoperiod at temperatures of 25 °C/18 °C (day/night). Tomato plants in the first generation (T_0_) came directly from tissue culture, and plants of the second generation (T_1_) were derived from seeds harvested from T_0_-generation transgenic plants. The flesh firmness of T_1_-generation tomato fruits was determined immediately after sampling. The samples were picked at four developmental stages; mature green (MG), breaker stage (BR), yellow red stage (YR), and red ripening stage (RR) based on the study of Liu et al. [44]. The remaining intact samples were immediately frozen in liquid nitrogen and stored at −80 °C for further analysis.

### 4.2. Measurement of Flesh Firmness in Grape Berries

The firmness was measured using a TA.XT plus texture analyzer (Stable Micro Systems, Surrey, UK), a P/2 probe (2 mm diameter), and a puncture method with a puncture depth of 7 mm. Fifteen berries were randomly selected for each biological replicate, and two parts above and below the maximum transverse diameter of each fruit were measured and averaged. There were three biological replicates for each of the five fruit stages.

### 4.3. Measurement of Water-Soluble Pectin (WSP) and Acid-Soluble Pectin (ASP) Contents

The test used was the carbazole colorimetry method described by He et al. (2018) [45]. The standard curve was drawn with the amount of galacturonic acid as the abscissa and the absorbance value as the ordinate. One-gram samples of grape berry and tomato fruit flesh were used to prepare the extracts. The WSP was extracted with distilled water in a 50 °C water bath, and the ASP was extracted with 0.5 mol/L sulfuric acid solution in a boiling water bath. The absorbances of 1 mL samples of the WSP and ASP extracts were measured using a spectrophotometer at 530 nm, and the concentrations were determined from the standard curve.

### 4.4. Genome-Wide Identification of VvPLs

The accession numbers and amino acid sequences of AtPLLs and SlPLs were collected using for grape *VvPLs* genes family analysis according to the previous study [30]. A total of 26 AtPL proteins and 22 SlPL proteins were used as queries to search the JGI database (https://phytozome-next.jgi.doe.gov/ (accessed on 26 November 2021)). The homologous protein alignment was performed using a Blastp research with 26 AtPLL and 21 SlPL protein sequences as the query sequence. All grape VvPL proteins are identified in the grape proteome.

### 4.5. Physicochemical Characteristics

The physicochemical characteristics was analyzed using the online tool ProtParam at expasy (http://web.expasy.org/protparam/ (accessed on 26 July 2021)) to calculate the number of amino acids, molecular weight, and isoelectric point (PI). The chromosome position was found via gene accession number in EnsemblPlants (http://plants.ensembl.org/index.html/ (accessed on 26 July 2021)). The amino acid sequences of VvPL family members were submitted into Plant-mPLoc server (http://www.csbio.sjtu.edu.cn/bioinf/plant-multi/ (accessed on 27 July 2021)) to predict the subcellular localization. The chromosomal location analysis was performed according to chromosomal positional information via online MG2C (http://mg2c.iask.in/mg2c_v2.1/ (accessed on 28 July 2021)).

### 4.6. Phylogenetic Tree Analysis

Twelve *OsPLLs* gene family numbers were collected from Zheng et al. (2018) [31]. All protein sequences of PL family members were collected from *Arabidopisis thaliana*, *Solanum lycopersicum*, *Oryza sativa* and *Vitis vinfera*, and used to construct phylogenetic tree via the MEGA7 software. The multiple sequence alignment result was obtained by the built-in Cluster W with default parameters. The construction method was selected Neighbor-joining and 1000 bootstrap replicates. The phylogenetic tree was modified iToL online tool (https://itol.embl.de/ (accessed on 1 August 2021)).

### 4.7. Conserved Motif, Gene and Protein Structure Analysis

The conservative model analysis was conducted via the online tool MEME (https://meme-suite.org/meme/tools/meme/ (accessed on 1 August 2021)) with the following parameters: maximum number of motifs, 10; and optimum motif width, between 6 and 50. Additionally, the result file was visualized via TBtools. The *VvPL* gene structure was analyzed to understand the gene characterization of transcription factors via the visualization server GSDS2.0 (http://gsds.gao-lab.org/index.php/ (accessed on 5 August 2021)). The three-dimensional structures of VvPL proteins were predicted using Phyre2 (http://www.sbg.bio.ic.ac.uk/phyre2/html/page.cgi?id=index/ (accessed on 11 August 2021)) and 3D model visualization was performed according to protein homology modeling by PyMoL.

### 4.8. Collinearity Analysis

The whole genome files of grape and Arabidopsis were obtained on the website (http://plants.ensembl.org/index.htm/ (accessed on 17 February 2022)), and the collinearity between grape and Arabidopsis was analyzed using the TBtools software [46].

### 4.9. Promoter Analysis

The 2000-bp sequences upstream of transcription factor of *VvPL* gene were used as the promoter sequence. The promoter cis-acting elements were predicted using the online site PlantCARE (http://bioinformatics.psb.ugent.be/webtools/plantcare/html/ (accessed on 10 August 2021)).

### 4.10. RNA Extraction and Quantitative Real-Time PCR (qRT-PCR)

Total RNA was extracted using the TIANGEN kit (Polysaccharides & Polyphenolics-rich; Tiangen Biotech, Beijing, China). cDNA was synthesized using a reverse transcriptase kit (Takara, Dalian, China). Primers for real-time quantitative PCR (RT-qPCR) were available in qPrimerDB-qPCR Primer Database (https://biodb.swu.edu.cn/qprimerdb/ accessed on 6 September 2021) and *VvActin* was used as a reference gene. Gene expression analysis was performed using system (BIO-RAD iQ5, Hercules, CA, USA) by three independent biological replicates for each sample. The relative expression level was calculated using the 2^−ΔΔCt^ method. The specific primers are listed in Appendix A.

### 4.11. Vector Construction and Plant Transformation

The coding sequence of *VvPL15* was cloned using kyoho grape flesh CDNA as a template. To get the transgenic lines, the coding sequence of *VvPL15* was inserted into pRI101 vector, named as *35S::VvPL15-OX*. *35S::VvPL15-OX* transgenic Arabidopsis, was obtained using the *Agrobacterium tumefaciens* strain GV3101 transformation method following the steps described in Kang et al. (2021) [47]. *35S::VvPL15-OX* transgenic tomato was obtained using the cotyledon disc method, as previously described [48]. The specific cloning primers were designed via the CE Design software (Appendix A).

### 4.12. Transient Transformation in Grape Leaves

The TRV vector was used to construct an antisense expression vector [49]. The specific 300 bp fragments of *VvPL15* were inserted into TRV vector in the antisense orientation, named as *TRV-VvPL15*. TRV1 was used as an auxiliary carrier. The antisense vectors were transformed using the *Agrobacterium tumefaciens* strain GV3101. The empty TRV vector served as a control. The detailed experimental operations followed the method of Kang et al. (2020) [50]. The specific cloning primers are listed in Appendix A. The gene expression, WSP and ASP contents of infected grape leaves were analyzed. Three biological replicates were designed, and each biological replicate was formed by mixing three grape leaves.

### 4.13. Statistical Analysis

Statistical differences were determined using single factor analysis of variance (ANOVA), and the least significant difference (LSD) at *p* < 0.05 by SPSS 26. The figures were prepared using GraphPad Prism9.

## 5. Conclusions

In this study, we identified and characterized the 16 members of the pectin lyase gene family in grape. Analysis of firmness and pectin content during grape berry ripening showed that changes in firmness were closely related to pectin composition. Analysis of gene expression patterns in berries from five stages of grape ripening revealed that *VvPL15*, a core member of the *VvPL* family, may play a major role in the onset of veraison in grapes. The role of *VvPL15* in grape ripening was investigated using heterologous transformation and transient transformation experiments, and we found that the relative expression levels of *VvPL15* promoted the softening of grape flesh through the depolymerization of pectin.

## Figures and Tables

**Figure 1 ijms-24-09318-f001:**
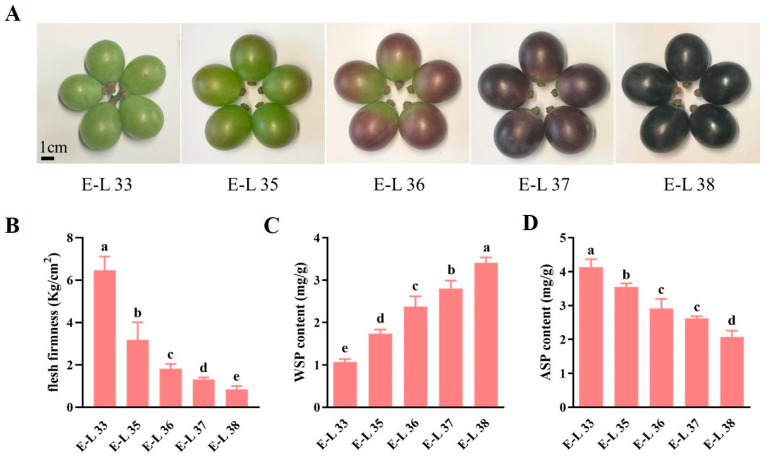
Determination of pectin content in ‘Kyoho’ grape fruit flesh during the ripening and softening. (**A**) The photogram of grape berries at five development stages. (**B**) The changes of firmness in grape flesh at five developmental stages. (**C**) The changes of WSP content. (**D**) The changes of ASP content. Values are mean ± SE, based on three independent biological replicates. Letters in figure indicate significant differences between groups (*p* < 0.05, one-way ANOVA).

**Figure 2 ijms-24-09318-f002:**
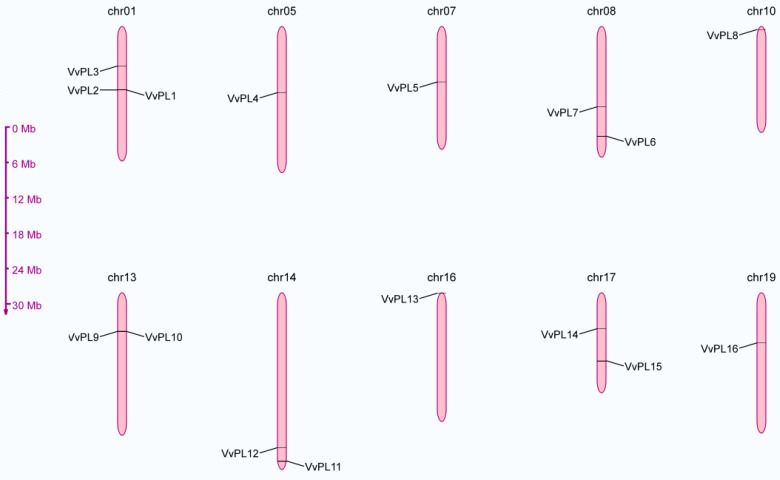
Chromosome location of VvPLs. chr: chromosomal.

**Figure 3 ijms-24-09318-f003:**
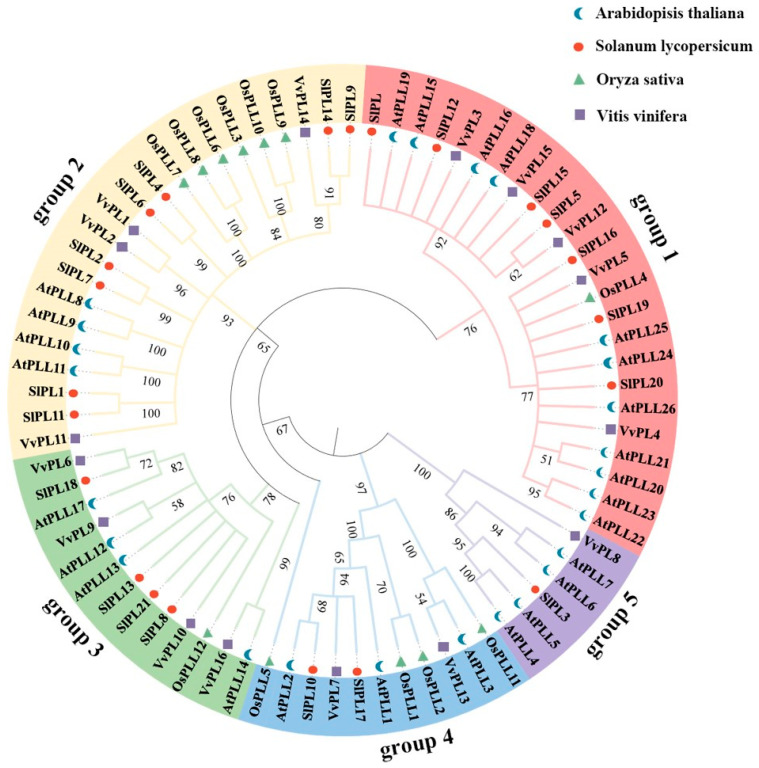
Phylogenetic tree of VvPLs from grape and other species. All PL proteins were divided into five groups, and different groups were labelled and distinguished with each other by different branch color. Furthermore, PL proteins from different species were labelled with different shape types.

**Figure 4 ijms-24-09318-f004:**
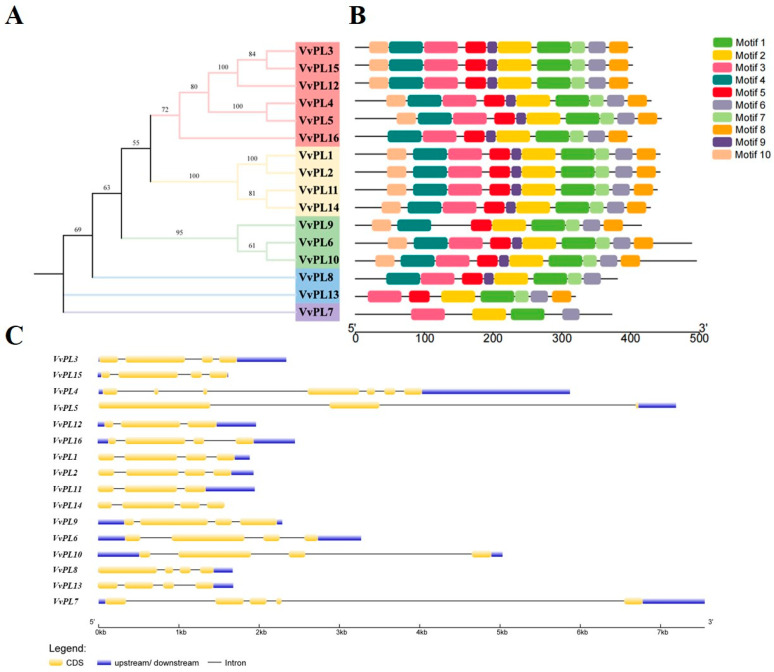
Phylogenetic relationship, motif compositions and upstream/downstream regions of VvPLs. (**A**) The unroot phylogenetic tree of VvPL proteins. Different colors represent different groups. (**B**) The distribution of conserved motif within each VvPLs. Boxes in different colors represent different conserved motifs, and their relative position are displayed. (**C**) The coding sequences (CDS) and untranslated regions (UTR) for *VvPLs* in grape. CDS and UTR are represented by yellow and blue boxes, respectively. The relative position is proportionally displayed based on the kilobase scale at the bottom of the figures.

**Figure 5 ijms-24-09318-f005:**
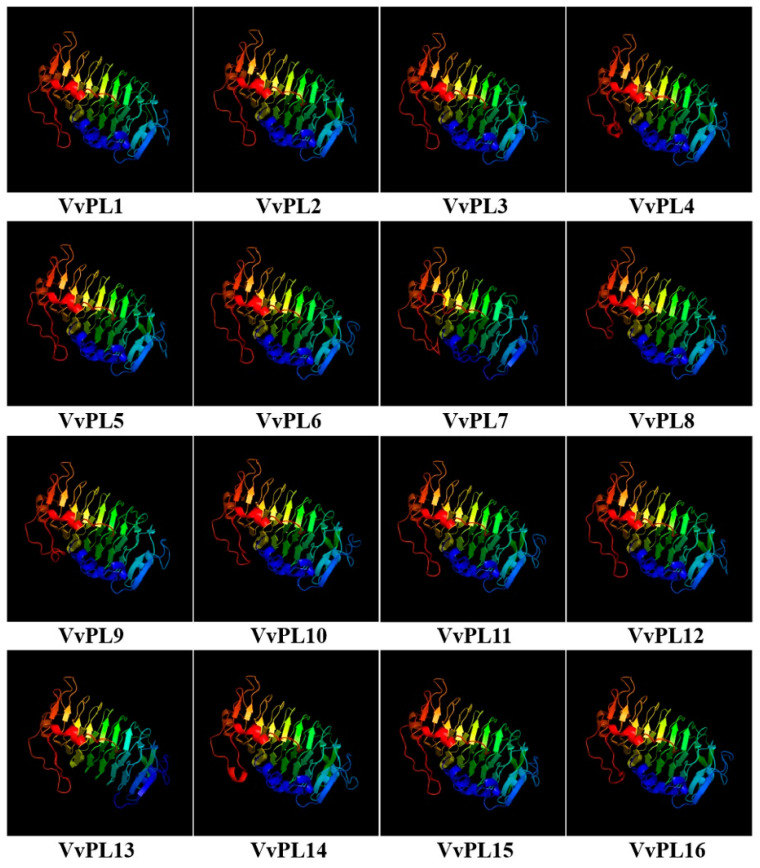
Tertiary structure of VvPLs. Predicted tertiary structure of the 16 VvPL proteins in grape. The protein structures all have the same domain color schemes. Structures reveal a high degree of structural homology in most gene members.

**Figure 6 ijms-24-09318-f006:**
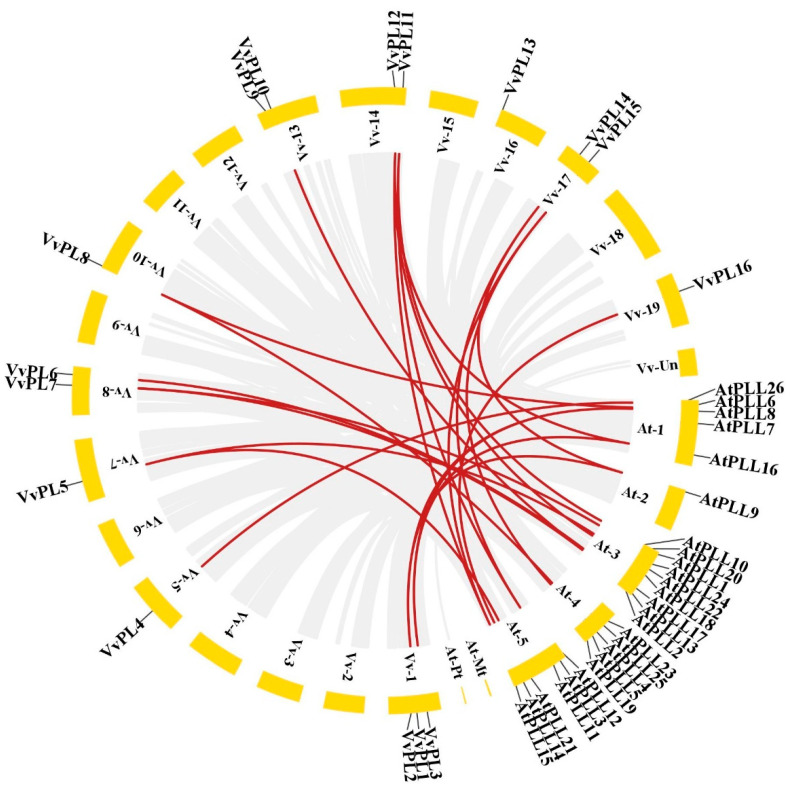
Collinearity analysis of the *VvPLs*. The collinear correlation for all genes of *PL* between grape and Arabidopsis is displayed. The localization of chromosomes was shown for grapevine and Arabidopsis in yellow and the collinear correlation was shown in red line.

**Figure 7 ijms-24-09318-f007:**
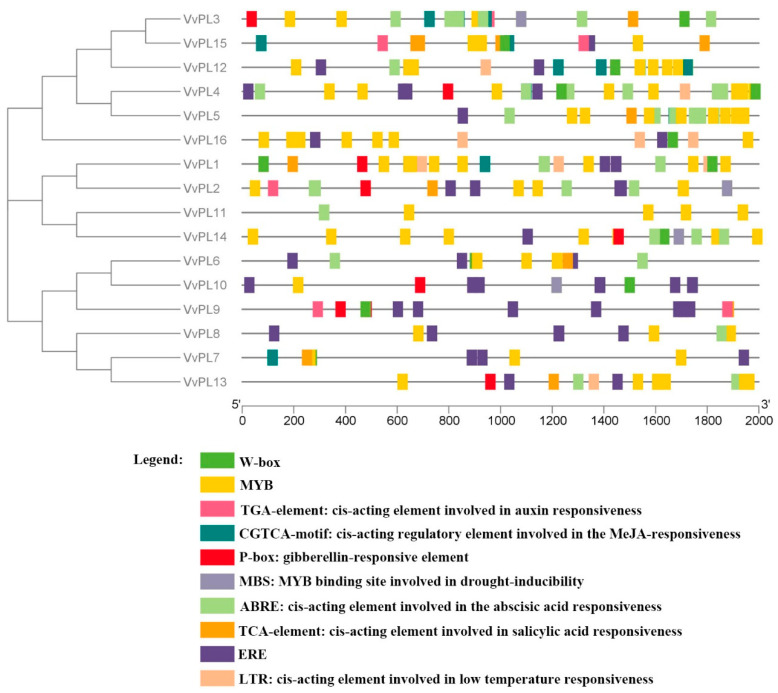
Putative cis-elements in *VvPL* gene promoters. Distribution of cis-elements on the promoter of *VvPLs*. Different color bars represent different cis-elements. The x-axis indicates the length of promoter sequence.

**Figure 8 ijms-24-09318-f008:**
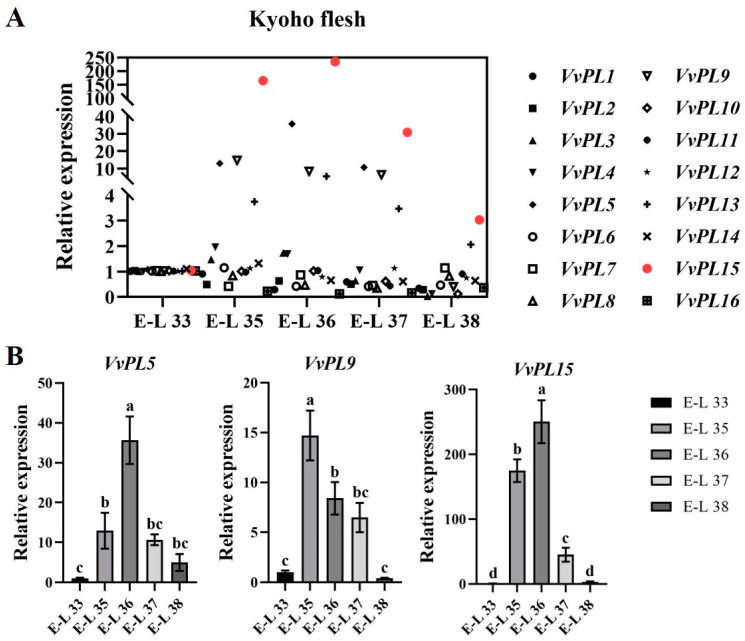
Expression pattern analysis of *VvPLs* in grape fruit flesh during ripening. (**A**) qRT-PCR analysis of 16 *VvPLs*. Red indicated highly expressed genes. (**B**) Columnar diagram of candidate *VvPLs*. *VvActin* served as a reference gene. Each value represents the means ± SE of three independent biological replicates. Significant differences (*p* < 0.05) between means are indicated by different letters.

**Figure 9 ijms-24-09318-f009:**
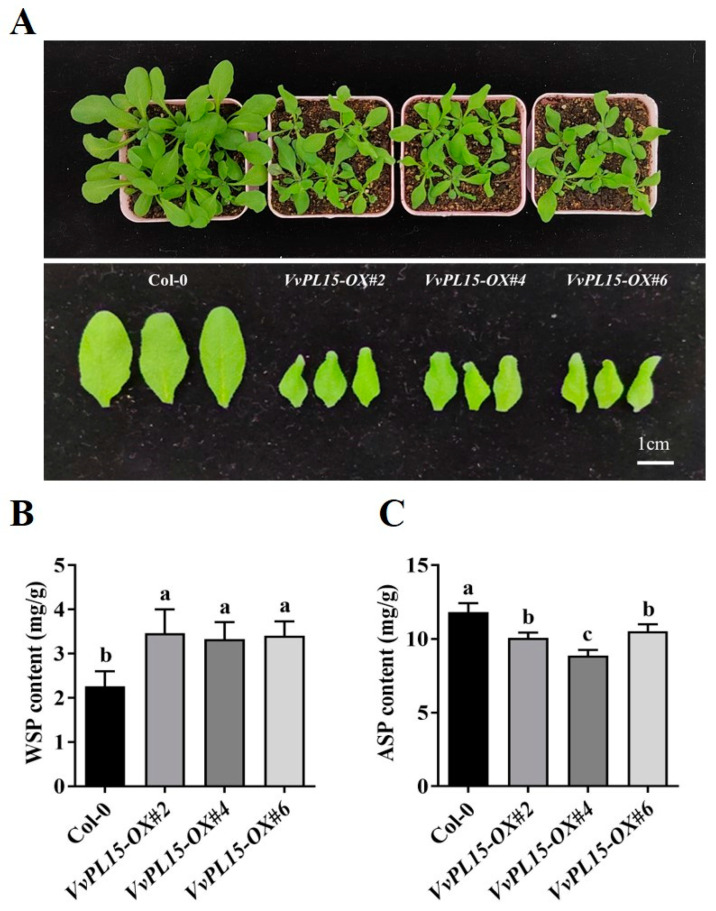
Phenotypic analysis of *VvPL15* overexpression in Arabidopsis. (**A**) Leaf phenotype of the *VvPL15-OX* and Col-0. (**B**) Comparison of WSP content between *VvPL15-OX* transgenic lines and Col-0. (**C**) Comparison of ASP content. Each value represents the means ± SE of three independent biological replicates. Significant differences (*p* < 0.05) between means are indicated by different letters.

**Figure 10 ijms-24-09318-f010:**
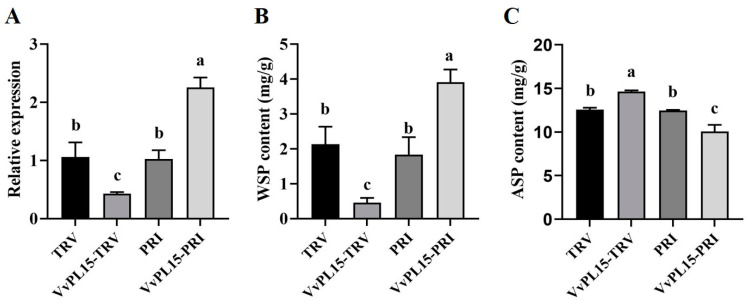
*VvPL15* affect pectin content. (**A**) Expression levels of *VvPL15* in TRV-, VvPL15-TRV-, PRI-, and VvPL15-PRI-infiltrated grape leaves two days after incubation. (**B**,**C**) Determination of WSP and ASP contents in TRV-, VvPL15-TRV-, PRI-, and VvPL15-PRI-infiltrated grape leaves two days after incubation. Bars with different letters are significantly different (*p* < 0.05) according to ANOVA. Data are shown as mean ± SE, based on three independent biological replicates.

**Figure 11 ijms-24-09318-f011:**
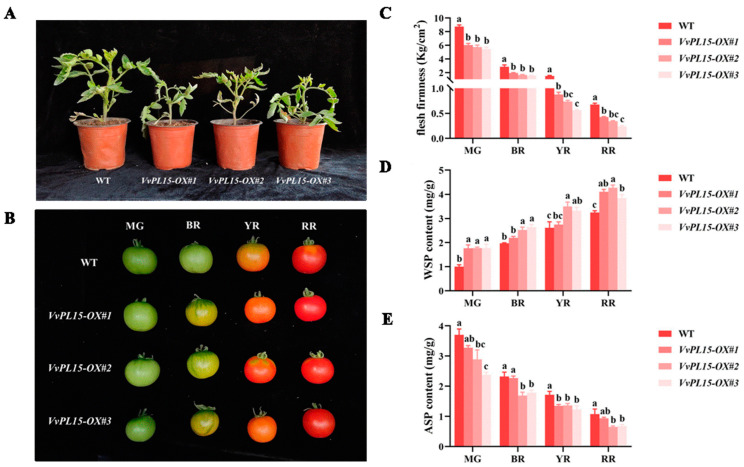
Phenotypic analysis of *VvPL15* overexpression in tomato. Phenotypic changes of WT and *VvPL15-OX* lines at seedling stage (**A**) and during fruit developing (**B**). (**C**) The changes of firmness in tomato flesh at four developmental stages. (**D**) The changes of WSP content. (**E**) The changes of ASP content. Values are mean ± SE, based on three independent biological replicates. Letters in figure indicate significant differences between groups (*p* < 0.05, one-way ANOVA).

**Table 1 ijms-24-09318-t001:** VvPL gene family with their molecular details.

Gene Name	Genome v2.1	Gene Length (bp)	Chromosome Location	Amino Acid Length (aa)	Molecular Weight (kDa)	Theoretical pI	Subcellular Localization
*VvPL1*	VIT_01s0026g01670	1888	chr01:10843187-10845074	443	49.93	9.44	Cell wall. Chloroplast.
*VvPL2*	VIT_01s0026g01680	1936	chr01:10846297-10848232	443	49.93	9.66	Chloroplast. Mitochondrion.
*VvPL3*	VIT_01s0137g00240	2338	chr01:6744527-6746440	403	44.33	6.49	Cell membrane. Chloroplast. Nucleus.
*VvPL4*	VIT_05s0051g00590	5871	chr05:11375744-11380850	430	46.98	8.01	Cell wall. Golgi apparatus. Mitochondrion.
*VvPL5*	VIT_07s0005g05520	7192	chr07:9546676-9551518	445	49.64	7.99	Chloroplast.
*VvPL6*	VIT_08s0007g04820	3276	chr08:18784751-18787240	489	53.84	6.00	Cell membrane. Cell wall. Chloroplast.
*VvPL7*	VIT_08s0040g02740	7549	chr08:13772697-13781967	373	41.46	8.16	Cell wall. Chloroplast. Cytoplasm. Golgi apparatus. Mitochondrion. Nucleus.
*VvPL8*	VIT_10s0116g01160	1676	chr10:603711-605197	381	42.37	9.06	Cell membrane. Cell wall. Chloroplast.
*VvPL9*	VIT_13s0019g04900	2294	chr13:6593309-6603253	416	45.57	6.10	Cell membrane.
*VvPL10*	VIT_13s0019g04910	5043	chr13:6603765-6608486	496	53.90	5.75	Cell membrane. Cell wall. Chloroplast. Nucleus.
*VvPL11*	VIT_14s0108g00030	1957	chr14:28869048-28870986	439	49.28	9.21	Cell membrane. Cell wall. Chloroplast. Cytoplasm. Golgi apparatus. Nucleus.
*VvPL12*	VIT_14s0219g00230	1973	chr14:26481825-26483236	403	44.44	6.88	Cell membrane. Cell wall. Chloroplast.
*VvPL13*	VIT_16s0039g00260	1689	chr16:133249-134937	320	36.10	9.00	Cell wall.
*VvPL14*	VIT_17s0000g05740	1581	chr17:6216007-6217587	429	47.88	8.48	Cell wall. Chloroplast.
*VvPL15*	VIT_17s0000g09810	1626	chr17:11675130-11676755	403	44.09	6.48	Cell membrane. Cell wall. Chloroplast. Nucleus.
*VvPL16*	VIT_19s0014g00510	2458	chr19:8585325-8587296	402	44.46	4.98	Cell membrane. Cell wall. Chloroplast.

## Data Availability

The original data for this present study are available from the corresponding authors.

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
