# Peer review of "VvPL15 Is the Core Member of the Pectate Lyase Gene Family Involved in Grape Berries Ripening and Softening"

_ijms, 2023, doi:10.3390/ijms24119318_

Round 1

Reviewer 1 Report

Aimed at understanding the role of pectin lyases (PLs) in fruit ripening in grapevine (Vitis vinifera), the authors performed phylogenetic analysis of PL genes using cultivar “Kyoho”. The authors were able to identify a total of 16 PL genes, namely VvPL 1 – 16, from grapevine. qRT-PCR revealed that VvPL5, VvPL9, and VvPL15 were significantly upregulated during fruit ripening in grapevine. With a focus of VvPL15, which showed the most dramatic change in its expression level during fruit ripening, the authors examined how VvPL15 regulates the content of water-soluble pectin (WSP) and acid-soluble pectin (ASP) in Arabidopsis and grape. Arabidopsis plants overexpressing VvPL15 showed increased WSP but reduced ASP in their leaves. Similarly, in grapevine leaves, transient overexpression of VvPL15 conferred increased WSP and reduced ASP, while transient suppression of VvPL15 conferred reduced WSP and increased ASP. These observations were consistent with previous findings on the role of PL genes in other plant species. Overall, the manuscript was well written. The finding that VvPL15 regulates WSP and ASP content in fruit ripening is a useful addition to the current knowledge on the functions of PLs. However, I have two major concerns about this manuscript.

1. Inadequate literature review. A recent paper by Li et al. (2023; https://www.mdpi.com/2311-7524/9/2/182) reported identification and functional analysis of VvPL genes. The authors should have compared their findings with the results reported by Li et al. (2023).  

2. Missing direct evidence on the effects of VvPL15 on fruit ripening. Two parameters, WSP and ASP, were used to indicate the progression of fruit ripening. Yet, fruit ripening is a sophisticated biological process involving a lot of changes. I was curious if the authors examined additional parameters, such as cell wall remodeling, during fruit softening. In addition, the authors should consider producing stable transgenic grapevine plants and/or other plants and testing if accelerated ripening can be conferred by VvPL15 overexpression (or if delayed ripening can be conferred by VvPL15 suppression).

Reviewer 2 Report

The ms contains numerous pitfalls that should be addressed

Abstract.

-          Avoid the use of abbreviations in the abstract (WSP, ASP).

-          What is the meaning of wild type leaf at maturity? (line 24)

Introduction:

-          The loss of cell turgor is another important factor in fruit softening, particularly in some fruits like grape. However, this process is not mentioned

-          Line 51. Pectins are erroneously defined. Pectins are complex polysaccharides composed mainly of three domains, HG, RGI and RGII.

-          Previous papers studied pectate lyase genes in grape (reference 46), but these are not included in the introduction

M&M:

-          The berries developmental stages should be defined (line 87)

-          What is the meaning of intelligent glasshouse in line 97?

-          Have been water-soluble and acid-soluble pectins extracted from cell wall material? This was not indicated in M&M section 2.3

-          Which genome was used for the identification of VvPls? (section 2.4)

-          VvPL15 silencing was achieved by antisense expression using VvPL15 fragments. This should be described in detail

-          According to section 2.13, mean separation was performed by LSD; however, the figure captions indicate that this was performed by Tukey test

Results

-          Some parts of the results should be included in the introduction or the discussion sections. For instance, the comment on lines 180-183 is an introductory text. Lines 187-189, 191-193, 252-253, 264-265, 283-287… should be in the discussion. Avoid the discussion of the results if the results and discussion are in separate sections

-          The phylogenetic tree contains PL genes from a few species

-          No conclusion can be deduced from Figure 8A since most genes are expressed at very low levels. Use a different scale for these low expressed genes.

-          It is concluded that VvPL15 has a key role in fruit softening. The major change in fruit firmness occurs in the transition from EL33 to EL35 stage; however, the gene is mainly expressed in EL35 and EL36. If VvPL15 would have a role in softening, an earlier expression of the gene could be expected

-          Another important question, is the expression of this gene fruit specific? Silencing experiments have been performed in grape leaves but they should have been done in fruit

Discussion: this section does not include a significant discussion of the results obtained and should be rewritten

Round 2

Reviewer 1 Report

I appreciate the efforts that the authors have made in responding to my previous concerns and questions.  The authors’ point-by-point responses have clarified all of the points I raised and strengthened the manuscript.